# Assessing Contact Time and Concentration of *Thymus vulgaris* Essential Oil on Antibacterial Efficacy In Vitro

**DOI:** 10.3390/antibiotics12071129

**Published:** 2023-06-29

**Authors:** Michela Galgano, Francesco Pellegrini, Daniela Mrenoshki, Paolo Capozza, Ahmed Hassan Omar, Anna Salvaggiulo, Michele Camero, Gianvito Lanave, Maria Tempesta, Annamaria Pratelli, Alessio Buonavoglia

**Affiliations:** 1Department of Veterinary Medicine, University Aldo Moro of Bari, Sp Casamassima Km 3, Valenzano, 70010 Bari, Italy; michela.galgano@uniba.it (M.G.); francesco.pellegrini@uniba.it (F.P.); daniela.mrenoshki@uniba.it (D.M.); paolo.capozza@uniba.it (P.C.); ahmed.omar@uniba.it (A.H.O.); anna.salvaggiulo@uniba.it (A.S.); michele.camero@uniba.it (M.C.); gianvito.lanave@uniba.it (G.L.); maria.tempesta@uniba.it (M.T.); 2Department of Biomedical and Neuromotor Sciences, Dental School, Via Zamboni 33, 40126 Bologna, Italy; alessio.buonavoglia85@gmail.com

**Keywords:** Gram-positive, Gram-negative, *Thymus vulgaris* L. essential oil, contact time, bacterial growth

## Abstract

The overuse and misuse of antibiotics can pose the risk of spreading mutant strains that show antimicrobial resistance (AMR), with negative impacts on the management of bacterial infections and economic implications for healthcare systems. The research and development of natural antibacterial agents could be a priority in the next years to improve a number of effective antibacterial molecules and to reduce the AMR phenomenon and its development. The present study identified the most effective concentration and contact time of *Thymus vulgaris* L. essential oil (TEO) to obtain bactericidal effects in vitro against different Gram-positive and Gram-negative bacterial strains. Six clinically isolated (wild types) bacterial strains, (*Citrobacter freundii, Enterococcus feciorum, Proteus mirabilis, Acinetobacter cioffi*, *Pseudomonas putrefaciens* and *Klebsiella pneumoniae*) and two ATCCs (*Staphylococcus aureus* and *Streptococcus mutans*) were tested after 1 min, 3 min and 5 min of contact with TEO. The preliminary results on *S. aureus* after 24 h of incubation revealed a TEO concentration of 9.28 mg/mL (*w*/*v*) that completely inhibited bacteria growth, keeping cell viability. The total suppression of bacterial growth at all tested contact times was observed for all tested bacterial strains, and the results were confirmed after 48 h of incubation. Bacterial growth suppression was confirmed even with the presence of organic components. These preliminary results showed the in vitro antimicrobial efficacy of TEO against different Gram-positive and Gram-negative bacterial strains. Future studies are necessary to confirm the reproducibility of these results even on other strains and to define the exact molecular mechanisms of EOs in order to consider TEO as a valid alternative to classic antibiotic therapies and subsequently to reduce the occurrence of AMR.

## 1. Introduction

The healing power of plants and their extracts is historically recognized, albeit empirically, as a centuries-old medicament. Many essential oils (EOs) extracted from plant species are biologically active and have carminative, antispasmodic, antitussive, expectorant, secretomotor, anthelmintic and bactericidal properties [1] However, with the advent of more effective powerful/potent synthetic antibiotics, the use of plant derivatives as antimicrobials has dramatically reduced. Consequently, antibiotics have largely been employed for therapeutic and prophylactic purposes in human medicine, animal husbandry and agriculture, but their indiscriminate and irrational use has determined the rapid onset of antimicrobial resistance (AMR). In recent years, the emergence of AMR to antibiotics has led to a trend reversal, and research into new (or old) natural drugs has become a priority worldwide. AMR has a global impact on public health with economic implications for healthcare systems and dramatically reduces the effectiveness of antibiotics treatments. AMR is associated with failure in bacterial infections, with prolonged hospitalization periods and with higher treatment costs and morbidity [2]. It should also be highlighted that AMR has led to the spread of multidrug resistance (MDR). *Pseudomonas aeruginosa*, *Staphylococcus aureus*, *Salmonella* spp., *Escherichia coli* and *Enterococcus* spp. are some of the most common and widespread multi-resistant bacteria responsible for community and hospital-acquired infections that are widely distributed worldwide. These infections are increasingly reported, confirming the ability of resistant bacteria to spread easily from the hospital environment to the community. Such resistant bacteria colonize healthy individuals even in the absence of risk factors that facilitate infections in fragile and elderly patients [3]. These bacteria represent a serious problem and a critical node for public health also due to potential interspecies transmission [2,4]. A link has been demonstrated between overuse and/or misuse at sub-therapeutic doses of antibiotics and the development of AMR [2,5,6], increasingly underlining the need for proper antibiotic management, reducing the use of antibiotics and the implementation of natural antibacterial agents. In this context, EOs, which are volatile oils or products of the secondary metabolism of aromatic plants, and the extracts of many plant species represent powerful tools to reduce the use of antibiotics and subsequently the spread of AMR in health-related areas and in the food industry [7,8,9]. Some of these EOs, largely employed in phytotherapy, cosmetics, agricultural pharmaceuticals and food industries, are plant extracts with demonstrated biological properties, such as anti-inflammatory, sedative, antioxidant, antiviral activities, and they are employed as substitutes for chemical-based preservatives [10]. In addition, several studies have demonstrated their antibacterial properties against pathogenic bacteria, including *E. coli*, *S. enterica*, *P. aeruginosa*, *S. aureus*, *Streptococcus epidermis*, *Klebsiella pneumoniae*, *Shigella* spp., *Proteus vulgaris* and *Bacillus cereus*, and above all, their ability to interfere with bacterial replication [11,12,13,14]. The main characteristic of EOs and their components is hydrophobicity. This lipophilicity allows fusion with the lipids of the cell membranes of bacteria and mitochondria and consequently favors the loss to a large extent of molecules and ions. The release of viable intracellular compounds (i.e., proteins, sugar and ATP) inhibits ATP generation, thus leading to the activation of a series of cascade reactions involving the entire cell and leading to the disruption of cell integrity, resulting in bacteria death [4]. To date, the antimicrobial activity of EOs is considered quite variable, being conditioned by the chemical composition and/or the concentration of their main components, which can act on the vital components of the bacteria (i.e., the lipid bilayer of the cell membrane) or on the main metabolic pathways (i.e., affecting the cell cycle and S phase and inhibiting protein synthesis and DNA replication) [15]. Several species of Gram-negative bacteria are affected in the mechanisms of the regulation of efflux pumps by some components of EOs [16]. The use of EOs in synergistic combinations with commonly used drugs or disinfectants can also allow reductions in the concentrations of both compounds, with the potential to improve the outcome of therapy and reduce collateral effects [17,18]. Among all the components of TEO, a pivotal role for antimicrobial activity is played by phenolic components (i.e., carvacrol, thymol, etc.), which are more effective when used in their entirety compared to single components, confirming the synergistic effect of the phenolic molecules of TEO [19]. EOs used in combination with antibiotics or as an alternative to antibiotics can counteract the emergence of AMR, prolonging the effectiveness of actual antibiotics and/or favoring the reintroduction of drugs that have lost activity against MDR strains [20]. Among EOs, recent studies have demonstrated high levels of the antibacterial activity of *Thymus vulgaris* L. essential oil (TEO) [14,21]. Although the antimicrobial activity of the phenolic compounds carvacrol and thymol is not fully clarified, some studies have shown that they favor the disruption of the outer and inner membrane through interaction with membrane proteins and with some intracellular targets [22,23]. Interestingly, a sub-lethal dose of thymol is able to promote the accumulation of misfolded proteins in the outer membrane of *Salmonella enterica*, and to upregulate genes involved in the synthesis of outer membrane proteins [24]. To date, no standardized methods are available for the evaluation of the antimicrobial activity of EOs that are different from what happens for antibiotics, for which valid, approved and well-established tests are available [25]. The bactericidal activity of drugs and EOs is generally assessed with the MIC (Minimum Inhibition Concentration) and the MBC (Minimum Bactericidal Concentration) methods, which represent a relevant metric/assessment value to predict the outcome of a therapy applied to a patient [26,27]. Furthermore, knowing the pharmacodynamic parameters of a molecule makes it possible to optimize their dosage, and to improve the outcome of infections [28]. In this context, PAE (post antibiotic effect), i.e., the suppression of bacterial growth observed after the removal of the antimicrobial agent from the culture medium [29], can be used as a “new” pharmacodynamic parameter. To our knowledge, a limited number of studies on PAE have been published in relation to the use of EOs [30]. Furthermore, the evaluation of the antimicrobial activity of TEO after a contact time of minutes instead of hours could have greater transnationality in clinical applications and therefore could have important implications, such as the reduction in the dosage and the consequent reduction in therapeutic costs that would reduce the toxic effects. Last but not least, the advantage of counteracting the development of the AMR phenomena should be considered [31,32]. To this purpose, the present study aimed to evaluate the different concentrations of TEO that can have bactericidal effects on both ATCCs and field bacterial strains. The innovation of our experimentation is related to the contact time of TEO with the bacterial suspensions. The different Gram-positive and Gram-negative bacterial strains were in fact put in contact with several concentrations of TEO for 1 min, 3 min and 5 min in order to identify the most effective concentration capable of having a bactericidal effect after only a few minutes of contact.

## 2. Results

### 2.1. TEO Cytotoxicity on Cell Cultures

The TEO maximum non-cytotoxic concentration was evaluated in vitro with the Toxicology Assay Kit and was considered as the TEO concentration at which the viability of treated Madin Darby Bovine Kidney (MDBK) cells decreased by no more than 20% (CC20) with respect to the negative control. The CC20 value of TEO was assessed at a 1:100 dilution (*v*/*v*), corresponding to a concentration of 9.28 mg/mL (*w*/*v*) and calculated as the mean ± standard deviation (SD) of three experiments. In all the experiments, the DMSO did not show any effect on MDBK cells.

### 2.2. MDR Activity

The screening for the MDR activity of the six clinical samples employed in the present study showed a variable degree of resistance to the different molecules tested. Details of the isolated strains about antimicrobial resistance are provided in Table 1.

### 2.3. TEO Antibacterial Activity

Preliminary bactericidal results on *S. aureus* after 24 h of incubation revealed that TEO at concentrations of 18.56 mg/mL and 9.28 mg/mL (*w*/*v*) totally inhibited bacteria growth after 5 min of contact at room temperature (RT). A decrease in the antibacterial efficacy was observed when TEO was tested at concentrations ranging from 3.712 mg/mL to 0.464 mg/mL (*w*/*v*), corresponding to a dilution of 1:250 to 1:2000 (*v*/*v*), and specifically, *S. aureus* growth reduction ranged from 10^1^ to 10^3^ CFU/mL when TEO was tested at concentrations ranging from 3.172 to 0.928 mg/mL (*w*/*v*), corresponding to a dilution of 1:250 to 1:1000 (*v*/*v*) (Figure 1).

As reported in Figure 1, no antibacterial efficacy was observed when TEO was tested at a concentration of 0.464 mg/mL (*w*/*v*), corresponding to a 1:2000 dilution (*v*/*v*). All the results were confirmed when bacterial growth was evaluated after 48 h of incubation.

Having identified 9.28 mg/mL as an effective antibacterial TEO concentration (*w*/*v*) in the absence of cytotoxicity, the bactericidal activity was also evaluated on all following bacterial strains considered in a recent study: *C. freundi*, *E. feciorum*, *P. mirabilis*, *A. cioffi*, *P. putrefaciens*, *K. pneumoniae* (field strains) and *S. mutans* (ATCC 70061) after 5 min, 3 min and 1 min of contact at RT. The same effective concentration, 9.28 mg/mL (*w*/*v*), was also employed to evaluate TEO efficacy on *S. aureus* after 3 min and 1 min of contact at RT. After 24 h of incubation, the total suppression of bacterial growth was observed at all contact times for all tested bacteria (Figure 2). The results were confirmed when bacterial growth was evaluated after 48 h of incubation.

The bactericidal activity of TEO at the evaluated effective antibacterial concentration (9.28 mg/mL, *w*/*v*, corresponding to a 1:100 dilution, *v*/*v*) was also tested in the presence of organic materials. A suspension of 6% sheep erythrocites was mixted with the *S. aureus* suspension and TEO dilution for 1 min, 3 min and 5 min. After 24 h of incubation, the TEO bactericidal activity was also confirmed in the presence of organic material, and no bacterial growth was observed.

### 2.4. Data Analysis

The normality of the distribution was evaluated with the Shapiro–Wilk normality test (W = 0.6004, *p*-value = 0.0002752). Regarding the correlation between the concentrations of TEO and the CFU/mL, a statistically significant inverse correlation between the chosen concentration of TEO and the CFU/mL was observed. Overall, TEO significantly inhibited bacterial growth at the selected concentration of 9.28 mg/mL (*w*/*v*) for *C. freundi*, *E. feciorum*, *P. mirabilis*, *A. cioffi*, *P. putrefaciens*, *K. pneumoniae* and *S. mutans* (ATCC 70061) after 5 min, 3 min and 1 min of contact at RT (CFU = 0.00 for all tested strains, *p* < 0.05) (Figure 2).

## 3. Discussion

The evolution of antimicrobial resistance observed in the last decades is the end result of long-term selective pressure applied to bacteria as a consequence of the overuse and misuse of antibiotics prescriptions. The emergence of AMR not only demands the effective antibiotic stewardship of existing options but requires the development of new antimicrobials. From this perspective, EOs could represent a valid alternative to antimicrobial agents that are effective against many microorganisms [33,34]. The antimicrobial activity of EO compounds derives from their ability to alter the bacterial cell wall and/or the cell membrane, causing the release of lipopolysaccharides [35,36,37]. Furthermore, the targeted cells undergo different structural and metabolic alterations (i.e., changes in ATP balance, modifications of DNA, protein synthesis, intracellular pH and cytoplasmic coagulation and the inhibition of quorum sensing) [36,37]. EOs may also affect cellular growth regulation, nutritional balances and energy conversion in bacterial cells [38]. The properties of EOs can differ depending on the composition and proportion of different components that can exert additive, antagonistic or synergistic effects [39]. In the present study, the antimicrobial activity of different concentrations of TEO was evaluated after 1 min, 3 min and 5 min of contact at RT with suspensions of different bacterial strains. Bacterial growth was then evaluated after 24 h and 48 h of incubation. A preliminary study was conducted to determine the toxic effect of TEO under the same conditions with eukaryotic cells. The CC20 value of TEO at a concentration of 9.28 mg/mL (*w*/*v*) showed that the above concentration has no effect on cell cultures. Similar results were obtained in previous studies by Oliveira et al. [40] using higher concentrations (25–100 mg/mL), and no relevant toxic effects were observed on cell viability. These results could be related to the antioxidant and protective effect of TEO on cells [40,41], confirming their safety in veterinary and human medicine for disinfection. After determining the maximum non-cytotoxic concentration at a 1:100 (*v*/*v*) TEO dilution, corresponding to a 9.28 mg/mL (*w*/*v*) concentration, our results show that non-cytotoxic TEO concentrations completely inhibited the microbial growth of all tested bacterial strains (both ATCCs and clinical samples) after 1 min of contact, keeping cell viability. Several studies have evaluated the antimicrobial activity of EOs, reporting MIC values from 0.16 mg/mL to 10 mg/mL (*w*/*v*), but most of them evaluated the efficacy after a longer contact time (ranging from a minimum of 15 min to 48 h) than those tested in this study [42,43,44,45]. To our knowledge, this is the first study to evaluate TEO antimicrobial activity after a contact time of less than 5 min using very low TEO concentrations. Similar to this work, a previous study conducted on some bacterial species, including *S. mutans*, by de Oliveira et al. [40] demonstrated that TEO is able to reduce microbial growth after 5 min of exposure. However, the TEO concentration used in this study was 200 mg/mL (*w*/*v*), a value that is much higher than that tested in this study/work, which was 9.28 mg/mL (*w*/*v*). However, these data allow us to make some observations. First, we can state that, when the antimicrobial action of EOs is evaluated in the laboratory, it is not simple, and it is not always possible to compare results obtained from the application of different laboratory tests and different EO extracts [39]. Furthermore, our study demonstrates that TEO has a marked bactericidal action even when used at low concentrations (9.28 mg/mL *w*/*v*), suggesting its use for future applications in addition to medical devices. To improve the use of EOs and/or replace harmful drugs, the effective biological activities of EOs greatly expand the possibilities of their use for practical disinfection applications, reducing costs and potential toxic effects and, above all, expanding the plethora of natural substances that can be used as antibacterials by counteracting the development of AMR.

## 4. Materials and Methods

### 4.1. Thymus vulgaris EO

The pure EOs of *Thymus vulgaris* (TEO) were provided by Specchiasol S.r.l. (Bussolengo, VR, Italy) and were stored in a brown glass bottle at a temperature of 0–4 °C for the entire duration of the experiments. The concentration of the commercial packaging of TEO, estimated by Catella et al. [46], was equal to 928 mg/mL (*w*/*v*). The chemical composition of TEO was determined via gas chromatography/mass spectrophotometry (GC/MS) as previously reported [12]. Approximately 25 components were identified, representing 98.7% of the total constituents detected [12,21], and among these the main components were thymol (47.01%), o-cymene (19.64%) and γ-terpinene (8.83%).

### 4.2. TEO Cytotoxicity on Cell Cultures

TEO at a stock concentration of 928 mg/mL (*w*/*v*) was diluted in DMSO and subsequently in Dulbecco Minimal Essential Medium (DMEM), as described above. MDBK cells were cultured in DMEM, and TEO cytotoxicity was assessed via an XTT assay [47] using the In Vitro Toxicology Assay Kit (Sigma–Aldrich Srl, Milan, Italy) after exposing the cells to the following final TEO dilutions: 1:50, 1:100, 1:250, 1: 500, 1:1000 and 1:2000 (*v*/*v*), ranging in concentration from 0.464 to 18.560 mg/mL (*w*/*v*), for 72 h. Cytotoxicity was assessed by measuring the absorbance signal (optical density, OD), with a spectrophotometer. Untreated cells were used as a negative control and were considered at 0% cytotoxicity, and cells treated with equivalent dilutions of DMSO were used as the vehicle control. Cytotoxicity data were analyzed with a non-linear curve fitting procedure, and the goodness of fit was evaluated via a non-linear regression analysis of the dose–response curve. The maximum non-cytotoxic concentration was considered as the TEO concentration at which the viability of treated MDBK cells decreased by no more than 20% (CC20) with respect to the negative control [25]. All experiments were performed in triplicate.

### 4.3. Bacterial Strains

All experiments were performed using the following bacterial strains: *Citrobacter freundii*, *Enterococcus feciorum*, *Proteus mirabilis*, *Acinetobacter cioffi*, *P. putrefaciens* and *K. pneumoniae*, isolated from samples submitted to the bacteriology laboratory of the Department of Veterinary Medicine, University of Bari, Italy. The six clinical strains were isolated from different samples (skin lesion, blood, milk, eye, mucous membranes and oral and pharyngeal swabs) collected from animals (large ruminants, small ruminants, pets, horses, rabbits and turtles), characterized via morphological studies and identified by means of standard biochemical tests (API 20E and API 20 Staph System, BioMérieux, Craponne, France). All the strains were stored at −20 °C until use in the culture medium Tryptic Soy Broth (TSB) (Oxoid, Milan, Italy) with glycerol 20%. Two ATCC strains, *S. mutans* (ATCC 70061) and *S*. *aureus* (ATCC 43300), were also employed (Manassas, VA, USA). Bacterial suspensions for experimental studies were prepared by inoculating 200 µL of each microorganism in 3 mL of TSB and were then incubated for 24 h at 37 °C. For the experiments, 10^9^ CFU/mL of each 24 h culture was used, except for *S*. *mutans*, which, after 24 h of incubation, showed a titer of 10^7^ CFU/mL.

### 4.4. Antimicrobial Susceptibility

Eleven different antibiotics (Amoxicillin + Clavulanic acid, AMC, 30 μg; Ampicillin, AMP, 10 μg; Gentamicin, CN, 10 μg; Oxytetracycline, OT, 30 μg; Ceftriaxone, CRO, 30 μg; Enrofloxacyne, ENR, 10 µg; Moxyfloxacin, MOX, 5 μg; Docycycline, DO, 30 μg; Cephalexin, CL, 30 μg; Cefotaxime, CTX, 30 μg; Co-Trimoxazole, SXT, 25 μg) were used to investigate in vitro the antimicrobial activity of the six clinical samples employed in the present study using the disk diffusion method (DDM). The antibiotics were selected based on standardized therapeutic protocols available for infection caused by Gram-negative and Gram-positive and tested according to Clinical & Laboratory Standards Institute (CLSI) guidelines. The European Committee for Antimicrobial Susceptibility Testing (EUCAST) (http://www.eucast.org/clinical_breakpoints/ (accessed on 3 May 2023)) and/or the indications of CLSI (https://clsi.org/media/2663/m100ed29_sample.pdf (accessed on 3 May 2023)) were used for the interpretation of the test after incubation at 37 °C for 24 h. Based on EUCAST interpretative criteria, the isolate strains were categorized as susceptible (S) or resistant (R) [12]. *S. aureus* ATCC 43300 was used as a quality control.

### 4.5. TEO Antibacterial Activity

The preliminary antibacterial activity of TEO was carried out on *S*. *aureus* (ATCC strain 43300). TEO was diluted (1:10) in dimethyl sulfoxide (DMSO; Sigma-Aldrich, St. Louis, MO, USA) and was subsequently diluted in TSB. TEO dilutions of 1:100, 1:200, 1:500, 1:1000, 1:2000 and 1:4000 (*v*/*v*) were tested with the established *S. aureus* inoculum in TSB (10^9^ CFU/mL) for 5 min at RT, achieving the following TEO final dilutions: 1:50, 1:100, 1:250, 1:500, 1:1000 and 1:2000 (*v*/*v*), corresponding to concentrations in *w*/*v* from 18.560 to 0.464 mg/mL. Then, 1 mL aliquots of each suspension were diluted (ten-fold dilutions starting from 10**^−^**^1^ to 10**^−^**^9^) in TSB and cultured into plate count agar (PCA) plates. The positive control (bacterial suspension without TEO) was contextually diluted and plated as above. All cultured plates were incubated at 37 °C for 24 h and 48 h. All tests were performed in triplicate.

Having identified 1:100 as the highest dilution of TEO completely inhibiting *S. aureus* growth after 5 min of contact, all the strains, both ATCCs and clinical samples, were tested with the aforementioned TEO dilutions to evaluate the antibacterial activity after 1 min, 3 min and 5 min of contact at RT. Bactericidal activity was evaluated after 24 h and 48 h of incubation at 37 °C, as described above. The tests conducted on *S. mutans* were carried out by incubating the strain at 37 °C in an atmosphere with 5% CO_2_. All the experiments were performed in triplicate.

### 4.6. Data Analyses

The distribution normality was evaluated via the Shapiro–Wilk test. Student’s *t*-test for independent samples or a one-way analysis of variance (ANOVA) was performed, followed by a Bonferroni post hoc test. A *p*-value < 0.05 was considered statistically significant. Statistical analyses were carried out with GraphPad Prism v8.1.2 (Dotmatics, Boston, MA, USA).

### 4.7. TEO Antibacterial Activity in the Presence of Sheep Erythrocytes

The antibacterial activity of TEO was evaluated also in the presence of organic samples (i.e., sheep erythrocytes) that could alter the antimicrobial activity of TEO. Briefly, an aliquot of each mixture composed of a bacterial suspension and TEO diluted at 1:100 (*v*/*v*) was tested in the presence of 6% erythrocytes. After 1 min, 3 min and 5 min of contact at RT, 1 mL aliquots of each mixture were diluted as described above from 10**^−^**^1^ to 10**^−^**^9^ in TSB, cultured into PCA plates and incubated for 24 h and 48 h at 37 °C. Each test was performed in triplicate.

## 5. Conclusions

These results demonstrate the great effectiveness of TEO after short-time contact. The fast in vitro antibacterial activity and biocompatibility of TEO could be a relevant feature for the possible therapeutic topical applications on mucosae (oral, nasal, pharyngeal) and skin, where rapid antimicrobial effects are required to treat mucositis, wounds and dermatitis [48]. Future studies are still needed to confirm the reproducibility of these results using a larger collection of clinical strains, to define the molecular mechanisms underlying the synergistic activities of EOs with antibiotic molecules or with disinfectants and to reduce the occurrence of AMR and contribute to improving public health.

## Figures and Tables

**Figure 1 antibiotics-12-01129-f001:**
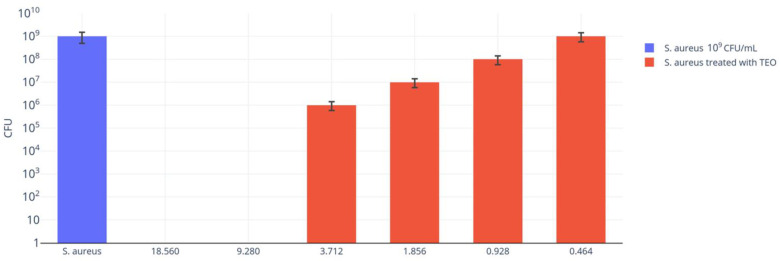
Antibacterial activity of TEO at different concentrations ranging from 18.560 mg/mL to 0.464 mg/mL (*w*/*v*) against *S. aureus* suspension after 5 min of contact. The starting concentration of *S. aureus* inoculum was 10^9^ CFU/mL in each mixture. Bacterial growth was evaluated after 24 h of incubation.

**Figure 2 antibiotics-12-01129-f002:**
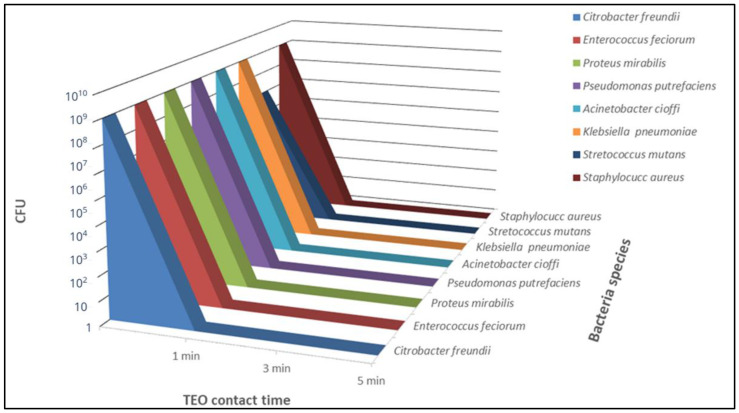
TEO antibacterial activity evaluated after 1 min, 3 min and 5 min of contact with different bacterial strains (ATCC and field strains). TEO was employed at a concentration of 9.28 mg/mL (*w*/*v*), corresponding to a 1:100 diution (*v*/*v*), and the bacteria inoculum concentration was 10^9^ CFU/mL in each mixture. Bacterial growth was evaluated after 24 h of incubation.

**Table 1 antibiotics-12-01129-t001:** Antimicrobial resistance profiles of the six isolated strains employed in the present study to test TEO activity.

Antibiotic	Bacterial Strains					
	*C. freundii*	*E. feciorum*	*P. mirabilis*	*P. putrefaciens*	*A. cioffi*	*K. pneumonie*
AMC	S	I	R	R	I	S
AMP	I	R	R	R	R	I
CN	S	S	I	I	I	I
OT	S	R	R	R	R	S
CRO	I	S	R	S	R	S
ENR	S	S	I	S	I	S
MOX	S	S	S	S	S	S
DO	S	I	R	R	I	S
CL	R	R	R	R	R	S
CTX	S	I	R	S	I	S
SXT	S	S	R	S	R	S

Legend: AMC: Amoxicillin + Clavulanic acid; AMP: Ampicillin; CN: Gentamicin; OT: Oxytetracycline; CRO: Ceftriaxone; ENR: Enrofloxacyne; MOX: Moxyfloxacin; DO: Docycycline; CL: Cephalexin; CTX: Cefotaxime; SXT: Co-Trimoxazole. S: Sensitive; I: Susceptible, Increased exposure; R: Resistant.

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
