# Peer review of "Assessing Contact Time and Concentration of Thymus vulgaris Essential Oil on Antibacterial Efficacy In Vitro"

_antibiotics, 2023, doi:10.3390/antibiotics12071129_

Round 1

Reviewer 1 Report

Do you compared the action of TEO with standard antibiotics?

MIC has been determined for bacterial strains that have been investigated?

Author Response

Reviewer 1

Do you compared the action of TEO with standard antibiotics?

MIC has been determined for bacterial strains that have been investigated?

REPLY: We have not compared TEO efficacy with standard antibiotics because the objective of the study is the evaluation of TEO activity after short contact time. Similarly, MIC was not evaluated because the data do not add relevant information to the purpose of the study. The strains were all antibiotics resistant, and the object is the use of TEO as an alternative to antibiotics.

Reviewer 2 Report

The authors have conducted a study on the bactericidal effect of an essential oil TEO. However, the authors have not described the mechanism of TEO in detail nor have presented any visual evidence. There have been a few studies on effectiveness of essential oils on bacteria without delving deep into the mechanism and therefore the impact of this study is moderate to low.

The authors must address the following questions

1) References required for applications mentioned in line 62-63.

2) Line 79-81. If EO's are used alone how can the effect be synergistic?Authors should make clear that there are multiple components in EO's even when they are used alone. Authors do this later in lines 87-88. Moving this information to the current line would make the sentence meaningful.

3) Line 130- It is not clear what authors mean by phenotype determinants.

4) It is usual to have a variations of ~0.5 log units in most antibacterial tests. Standard deviations/standard errors must be plotted on the graph clearly to show that the efficacies at different concentrations are truly different.In addition to this, authors need to perform a statistical significance test (student t-test) of the data in figure 1.

5)Line 185-187, The sentence is confusing and some part of it is redundant. It can be rephrased as "The emergence on AMR not only demands effective antibiotic stewardship of existing options but requires development of new antimicrobials. 

6) Line 270- In figure 2, the initial concentration of S. mutants is also depicted as 10^9 CFU/mL. However, authors here say it is 10^7 CFU/mL.

7) TEO-antibacterial activity section- What is the mixture? How much of bacteria is injected into each TEO dilution? The authors need to clearly describe the amounts/volumes.

What was arresting/stopping the effect of TEO on bacteria after the 1, 3 and 5 min of contact times? Typically a neutralizer solution such as Dey Engley broth or Letheen broth is used to stop the antimicrobial ingredient to nullify the effect. Why wasn't a neutralizer solution used to arrest the effect of TEO after the necessary contact times?

This is important as the time of action in case TEO antibacterial activity is not arrested will be longer than what authors claim a 1, 3 and 5 minutes. It will be as long as the incubation time where TEO continues to act on the agar plates.

8) High quality/resolution images of the control plates and plates inoculated are impactful with such kinds of studies.

Overall, the quality of English must be improved for a technical paper. There are instances where one can see spoken English in the sentences. This must be omitted in a technical paper. Moderate revision of English is required in the manuscript to provide clarity to the reader. Please refer to the attached file for suggestions.

Author Response

Reviewer 2

The authors have conducted a study on the bactericidal effect of an essential oil TEO. However, the authors have not described the mechanism of TEO in detail nor have presented any visual evidence. There have been a few studies on effectiveness of essential oils on bacteria without delving deep into the mechanism and therefore the impact of this study is moderate to low.

The authors must address the following questions.

1) References required for applications mentioned in line 62-63.

REPLY: Reference was added.

2) Line 79-81. If EO's are used alone how can the effect be synergistic? Authors should make clear that there are multiple components in EO's even when they are used alone. Authors do this later in lines 87-88. Moving this information to the current line would make the sentence meaningful.

REPLY: As suggested the sentence was moved.

3) Line 130- It is not clear what authors mean by phenotype determinants.

REPLY: The development of antibiotic resistance is usually associated with genetic changes, either to the acquisition of resistance genes, or to mutations in elements relevant for the activity of the antibiotic. However, in some situations resistance can be achieved without any genetic alteration; this is called phenotypic resistance. In our study we have not evaluated resistance genes, but only the antimicrobial susceptibility according to Clinical & Laboratory Standards Institute (CLSI) guidelines.  The European Committee for Antimicrobial Susceptibility Testing (EUCAST) (http://www.eucast.org/clinical_breakpoints/) and/or the indications of CLSI (https://clsi.org/media/2663/m100ed29_sample.pdf) were used for the interpretation of the test and based on EUCAST interpretative criteria, the isolate strains were categorized as susceptible (S), Susceptible, increased exposure (I), or resistant (R).

4) It is usual to have a variations of ~0.5 log units in most antibacterial tests. Standard deviations/standard errors must be plotted on the graph clearly to show that the efficacies at different concentrations are truly different. In addition to this, authors need to perform a statistical significance test (student t-test) of the data in figure 1.

REPLY: Figure 1 has been corrected as suggested by adding Standard Error. Moreover T-student test was performed on the dataset showing p value < 0.05 for all concentration, except 1:2000 (p value = 0.1103). All analysis were performed with R-studio software.

5)Line 185-187, The sentence is confusing and some part of it is redundant. It can be rephrased as "The emergence on AMR not only demands effective antibiotic stewardship of existing options but requires development of new antimicrobials. 

REPLY: The Authors are grateful to the Reviewer for the suggestion. The sentence was rephrased as suggested.

6) Line 270- In figure 2, the initial concentration of S. mutants is also depicted as 10^9 CFU/mL. However, authors here say it is 10^7 CFU/mL.

REPLY: The Authors are grateful to the Reviewer for this observation. There was a typo in Figure 2. The initial concentration of S. mutants is 10^7 CFU/mL. The figure was modified.

7) TEO-antibacterial activity section- What is the mixture? How much of bacteria is injected into each TEO dilution? The authors need to clearly describe the amounts/volumes.

REPLY: Reviewer is right, and the text could be misunderstood. The mixture is represented by TEO dilution and bacteria suspension. TEO was diluted 1:100, 1:200, 1:500, 1:1000, 1:2000, 1:4000 (v/v) in TSB. Then each TEO dilution (1mL) was added to 1mL of bacteria suspension. Therefore, the TEO final dilutions were 1:50, 1:100, 1:250, 1:500, 1:1000 and 1:2000 (v/v), corresponding to concentration in w/v from 18.560 to 0.464 mg/mL.

We have better clarify the sentence in the text.

7-bis)What was arresting/stopping the effect of TEO on bacteria after the 1, 3 and 5 min of contact times? Typically a neutralizer solution such as Dey Engley broth or Letheen broth is used to stop the antimicrobial ingredient to nullify the effect. Why wasn't a neutralizer solution used to arrest the effect of TEO after the necessary contact times?

This is important as the time of action in case TEO antibacterial activity is not arrested will be longer than what authors claim a 1, 3 and 5 minutes. It will be as long as the incubation time where TEO continues to act on the agar plates.

REPLY: We have not employed any stop solution in order not to alter the experiments and leave the solution to be tested composed only of TEO and bacteria. The stop solution could have antibacterial activity and therefore could invalidate the test. After the contact time, the TEO dilutions (starting from 10^1 to 10^8) were performed immediately (after 10-15 seconds). TEO dilutions were so extreme (up to 10^8) that TEO loses its effectiveness, i.e. already after 10^3 its activity is no longer evident and disappears.

8) High quality/resolution images of the control plates and plates inoculated are impactful with such kinds of studies.

REPLY: The Authors agree with the Reviewer. The choice not to include the images arose from the fact that a plate with bacterial growth and one without bacterial growth (as results of the action of the TEO of our study) we believe are not significant for the objective of the study.

Overall, the quality of English must be improved for a technical paper. There are instances where one can see spoken English in the sentences. This must be omitted in a technical paper. Moderate revision of English is required in the manuscript to provide clarity to the reader. Please refer to the attached file for suggestions.

REPLY: Authors are grateful to the reviewer for the revisions and suggestions. English was improved.

Reviewer 3 Report

In this study, authors tested the antimicrobial activity of TEO. They observed that even 1 minute of contact time caused the complete elimination of the tested bacterial species. Testing and understanding the antimicrobial activity of natural products are very important to combat the increasing burden of antimicrobial resistance. However, in this study, authors have not provided all the data which they mention in the text and haven’t given detail experimental procedures, and thus make this study difficult to understand. I have following concerns about this study.

1) Line 114: Authors mentioned that the innovation in their study is contact time. In this case, they should mention how they made sure TEO was completely removed before plating the TEO treated bacterial culture in the procedure section.

2) Line 121, first paragraph, MDBK cells: Data is not provided. Didn’t mention which tested concentrations of TEO were toxic or non-toxic to MDBK.

3) Line 132, Table 1: Raw data and statistical significance should be provided (either in main or supplementary data) for tested antibiotics. What are the criteria for intermediate susceptibility? They should describe it in the text. 

Authors did not mention the purpose of testing different antibiotics.

4) Line 138: Authors mention that "18,56 mg/mL and 9.28 mg/mL (w/v) TEO totally inhibit the bacterial growth". Does that mean not even a single colony was observed after treating 109 CFU/mL bacteria with TEO for five minutes? Representative raw data should be provided for this result.
Figure 1: The statistical significance of the result is not provided.
If TEO could be inactivated by heat or any other reasonable means, then author should test the bacterial culture with the inactivated TEO as a negative control. This will enhance their antimicrobial results.

5) Lines 158–160: It’s a confusing statement. Please mention the time point when results were captured.

Figure 2: This is an interesting result, as authors observed zero CFU after 1, 3, and 5 minutes of treatment. They should put representative raw data in the manuscript.

6) Line 176–181: Treatment of S. aureus with TEO in the presence of 6% sheep erythrocytes, data is not in the manuscripts. Authors should explain why they tested TEO antimicrobial activity in the presence of erythrocytes in the text.

Minors:

1) It would be helpful for reader if results are divided into different sub-sections.

2) Line 107: please correct Eos

3) Line 278: please correct Gram and Gram +

4) Please mention Gram positive and Gram negative or create an abbreviation section and mention it there. Also, give the full form of all abbreviations in the abbreviation section.

5) Line 242: "TEO at stock concentration of 928 mg/mL (w/v)" this statement is confusing. 

Required editing at many places. Some of the edits are mentioned in the comments.

Author Response

Reviewer 3

In this study, authors tested the antimicrobial activity of TEO. They observed that even 1 minute of contact time caused the complete elimination of the tested bacterial species. Testing and understanding the antimicrobial activity of natural products are very important to combat the increasing burden of antimicrobial resistance. However, in this study, authors have not provided all the data which they mention in the text and haven’t given detail experimental procedures, and thus make this study difficult to understand. I have following concerns about this study.

1) Line 114: Authors mentioned that the innovation in their study is contact time. In this case, they should mention how they made sure TEO was completely removed before plating the TEO treated bacterial culture in the procedure section.

REPLY: The Authors have not employed any stop solution in order not to alter the experiments and leave the solution to be tested composed only of TEO and bacteria. After the contact time, the TEO dilutions (starting from 10^1 to 10^8) were performed immediately (after 10-15 seconds). TEO dilutions were so extreme (up to 10^8) that TEO loses its effectiveness, i.e. already after 10^3 its activity is no longer evident and disappears.

2) Line 121, first paragraph, MDBK cells: Data is not provided. Didn’t mention which tested concentrations of TEO were toxic or non-toxic to MDBK.

REPLY: The Authors have not understood the comments. In Materials and Methods (paragraph 4.2. TEO cytotoxicity on cell cultures) the test is described. We would be very grateful if the Reviewer could specify the comment.

3) Line 132, Table 1: Raw data and statistical significance should be provided (either in main or supplementary data) for tested antibiotics. What are the criteria for intermediate susceptibility? They should describe it in the text. (KIT………….

Authors did not mention the purpose of testing different antibiotics.

REPLY: There is no statistical significance for tested antibiotics. The criteria for the evaluation of susceptibility was performed according to “Antibiogramma Susceptibility testing (Diagnostici Liofilchem, Teramo, Italy). Description: Antibiotic Disc are paper discs with special features, that are impregnated with antibiotic and used for the susceptibility test according to the Kirby-Bauer antibiotic testing (KB testing or disk diffusion antibiotic sensitivity testing). Antibiotic Disc are available in a large variety of configurations. The discs are applied to the surface of a culture medium inoculated with a pure colony suspension of the microorganism under examination. After incubation, the plates are examined, the inhibition halos around each disc are examined and compared with the standard inhibition haloes: in this way the microorganisms are defined as being susceptible, intermediate or resistant to the tested antimicrobial agents. At the end of the incubation period, measure the inhibition halos and interpret according to the current reference standards: https://www.liofilchem.com/images/prodotti-evidenza/antibiotic-disc/disc_interpretative_table.pdf

The authors wanted to test multiple antibiotics to demonstrate that the strains used in the work have a broad spectrum of antibiotic resistance.

4) Line 138: Authors mention that "18,56 mg/mL and 9.28 mg/mL (w/v) TEO totally inhibit the bacterial growth". Does that mean not even a single colony was observed after treating 109 CFU/mL bacteria with TEO for five minutes? Representative raw data should be provided for this result.

REPLY: The Authors confirm that not even a single colony was observed after treating 109 CFU/mL bacteria with TEO for five minutes. The Authors did not include raw data because, except for high TEO dilution, no colony was observed.

Figure 1: The statistical significance of the result is not provided.

REPLY: T-student test was performed on the dataset showing p value < 0.05 for all concentration, except 1:2000 (0.464, p value = 0.1103). Figure 1 has been corrected by adding Standard Error. All analysis were performed with R-studio software.

If TEO could be inactivated by heat or any other reasonable means, then author should test the bacterial culture with the inactivated TEO as a negative control. This will enhance their antimicrobial results.

REPLY: Reviewer is right, and several tests were performed. The authors tried to inactivate TEO with heat (three cycles in an autoclave, and by boiling). But TEO still showed antibacterial activity. Adding other substances did not seem appropriate because any added molecule could interfere with the action of TEO or be itself an antibacterial to some extent, compromising our tests. TEO is widely recognized as an antibacterial. The purpose of our work is to demonstrate that it can carry out its action after a few minutes of contact.

5) Lines 158–160: It’s a confusing statement. Please mention the time point when results were captured.

REPLY: The time point for the evaluation of the results are 24h and 48h. In the text this information is reported (lines 142: “Preliminary bactericidal results on S. aureus after 24h incubation, revealed that TEO at concentrations of 18,56mg/mL and 9.28mg/mL (w/v) totally inhibited bacteria growth after 5 min of contact at room temperature (RT).”. Lines 156: “All the results were confirmed when bacterial growth was evaluated after 48h incubation.”)

Figure 2: This is an interesting result, as authors observed zero CFU after 1, 3, and 5 minutes of treatment. They should put representative raw data in the manuscript.

REPLY: As no growth was observed we deemed it unnecessary to include raw data

6) Line 176–181: Treatment of S. aureus with TEO in the presence of 6% sheep erythrocytes, data is not in the manuscripts. Authors should explain why they tested TEO antimicrobial activity in the presence of erythrocytes in the text.

REPLY: The Authors have included the test with erythrocytes because organic materials could alter antibacterial activity. The paragraph 4.7. “TEO antibacterial activity in the presence of sheep erythrocytes” describes the procedure. The Authors have better specify the objective of this test: “The antibacterial activity of TEO was evaluated also in the presence of organic sample (i.e. sheep erythrocytes) that could alter antimicrobial activity of TEO. Briefly, an aliquot of each mixture composed of bacterial suspension and TEO diluted 1:100 (v/v), was tested in the presence of 6% erythrocytes. After 1 min, 3 min and 5 min of contact at RT, 1mL aliquots of each mixture was diluted as described above from 10-1 to 10-9 in TSB, cultured into PCA plates and incubated for 24h and 48h at 37°C. Each test was performed in triplicate).

Minors:

1) It would be helpful for reader if results are divided into different sub-sections.

REPLY: As suggested, Results section was divided into different sub-sections.

2) Line 107: please correct Eos

REPLY: The text was modified

3) Line 278: please correct Gram and Gram +

REPLY: The text was modified

4) Please mention Gram positive and Gram negative or create an abbreviation section and mention it there. Also, give the full form of all abbreviations in the abbreviation section.

REPLY: As suggested, Gram positive and Gram negative was mentioned all over the text. An abbreviation section was also created at the end of the manuscript.

5) Line 242: "TEO at stock concentration of 928 mg/mL (w/v)" this statement is confusing. 

REPLY: As suggested the statement was rephrased.

Round 2

Reviewer 2 Report

Thank you for clarifying questions and modifying the manuscript to provide clarity.

Only one suggestion in regards to neutralizing solution. Once the bacterial test is done neutralizing solutions are often employed to arrest the activity of antimicrobial additives especially when one is conducting time based studies. Letheen broth, Dey Engley broth are such neutralizing solutions commonly used against most antimicrobials only to arrest their activity and they do not affect the growth of bacteria in any way. The authors have clarified that the dilutions are performed immediately and therefore TEO activity is lost after a couple of dilutions. But in the future, for time-based studies, especially at shorter times in the order of minutes,  it is suggested that one uses a neutralizing solution. 

Author Response

Reviewer #2

Thank you for clarifying questions and modifying the manuscript to provide clarity.

Only one suggestion in regards to neutralizing solution. Once the bacterial test is done neutralizing solutions are often employed to arrest the activity of antimicrobial additives especially when one is conducting time based studies. Letheen broth, Dey Engley broth are such neutralizing solutions commonly used against most antimicrobials only to arrest their activity and they do not affect the growth of bacteria in any way. The authors have clarified that the dilutions are performed immediately and therefore TEO activity is lost after a couple of dilutions. But in the future, for time-based studies, especially at shorter times in the order of minutes,  it is suggested that one uses a neutralizing solution. 

Reply to Reviewer #2: The Authors are grateful to R#2 for the suggestions which improved the quality of the manuscript. Certainly, for our future time-based studies , the suggested solutions will be employed.

Reviewer 3 Report

In this study, Authors tested the antibacterial activity of TEO against different bacterial species. However, previously, many studies have shown the antibacterial activity of TEO on the same bacterial species. Authors observed inhibition of bacterial species even after a minute of treatment at higher concentrations. In this situation, cytotoxic data is important, and authors did not provide it again. They mention the MDBK cell line cytotoxicity check (lines 128–135), but the data is not in the manuscript. Similarly, data on TEO treatment in the presence of erythrocytes is not provided in the manuscript. Authors confirm that no CFU was observed even after one minute of treatment, but raw data is not provided for any of the results. They could have tested their assay by different biochemical ways which can detect live and dead bacteria.

Authors should provide some mechanism for antimicrobial activity.

Author Response

Reviewer #3

In this study, Authors tested the antibacterial activity of TEO against different bacterial species. However, previously, many studies have shown the antibacterial activity of TEO on the same bacterial species. Authors observed inhibition of bacterial species even after a minute of treatment at higher concentrations. In this situation, cytotoxic data is important, and authors did not provide it again. They mention the MDBK cell line cytotoxicity check (lines 128–135), but the data is not in the manuscript. Similarly, data on TEO treatment in the presence of erythrocytes is not provided in the manuscript. Authors confirm that no CFU was observed even after one minute of treatment, but raw data is not provided for any of the results. They could have tested their assay by different biochemical ways which can detect live and dead bacteria.

Authors should provide some mechanism for antimicrobial activity.

Reply to Reviewer #3: The Authors are grateful to R#3 for the suggestions which improved the quality of the manuscript. However, some requests of R#3 are described in the manuscript.

R#3 question: In this situation, cytotoxic data is important, and authors did not provide it again. They mention the MDBK cell line cytotoxicity check (lines 128–135), but the data is not in the manuscript.

Reply: The cytotoxicity check on MDBK cells is described in M&M (paragraph 4.2, lines 263-276) and in Results (paragraph 2.1, lines 130-136).

R#3 question: Similarly, data on TEO treatment in the presence of erythrocytes is not provided in the manuscript.

Reply: Data on TEO treatment in the presence of erythrocytes is reported in M&M (paragraph 4.7, lines 333-338) and in the Results section (lines 179-183).

R#3 question: Authors confirm that no CFU was observed even after one minute of treatment, but raw data is not provided for any of the results.

Reply: As reported in our previous rebuttal, the Authors did not include raw data because, except for high TEO dilution, no colony was observed. T-student test was performed on the dataset showing p value < 0.05 for all concentration, except 1:2000 (0.464, p value = 0.1103).

R#3 question: They could have tested their assay by different biochemical ways which can detect live and dead bacteria.

Reply: The test employed in the present study were only microbiological test that revealed bacterial growth (then live bacteria). The objective of the study is the evaluation of the antibacterial activity of TEO.

R#3 question: Authors should provide some mechanism for antimicrobial activity.

Reply: Reviewer is right and certainly, some mechanism for antimicrobial activity would enrich the study. Therefore, this is a time-based study on the efficacy of TEO as antimicrobial drug after few minutes of contact. In the light of the promising results obtained, we are working on the evaluation of the mechanisms of action in order to obtain a broader picture of the actions of this oil.